# E/D-Mode GaN Inverter on a 150-mm Si Wafer Based on p-GaN Gate E-Mode HEMT Technology

**DOI:** 10.3390/mi12060617

**Published:** 2021-05-27

**Authors:** Li-Fang Jia, Lian Zhang, Jin-Ping Xiao, Zhe Cheng, De-Feng Lin, Yu-Jie Ai, Jin-Chao Zhao, Yun Zhang

**Affiliations:** 1Laboratory of Solid State Optoelectronics Information Technology, Institute of Semiconductors, CAS, Beijing 100083, China; lfjia@semi.ac.cn (L.-F.J.); zhanglian07@semi.ac.cn (L.Z.); zhecheng@semi.ac.cn (Z.C.); dflin@semi.ac.cn (D.-F.L.); aiyujie@semi.ac.cn (Y.-J.A.); 2Center of Materials Science and Optoelectronics Engineering, University of Chinese Academy of Sciences, Beijing 100049, China; 3Silan Integrated Circuit Co., Ltd., Hangzhou 310018, China; xiaojinping@silanic.com.cn; 4Lishui Zhongke Semiconductor Material Co., Ltd., Lishui 323000, China; 5Xiamen Silan Advanced Compound Semiconductor Co., Ltd., Xiamen 361026, China; zhaojinchao@silanled.com

**Keywords:** AlGaN/GaN, E-mode, D-mode, p-GaN, inverter, monolithic integration, small variations

## Abstract

AlGaN/GaN E/D-mode GaN inverters are successfully fabricated on a 150-mm Si wafer. P-GaN gate technology is applied to be compatible with the commercial E-mode GaN power device technology platform and a systematic study of E/D-mode GaN inverters has been conducted with detail. The key electrical characters have been analyzed from room temperature (RT) to 200 °C. Small variations of the inverters are observed at different temperatures. The logic swing voltage of 2.91 V and 2.89 V are observed at RT and 200 °C at a supply voltage of 3 V. Correspondingly, low/high input noise margins of 0.78 V/1.67 V and 0.68 V/1.72 V are observed at RT and 200 °C. The inverters also demonstrate small rising edge time of the output signal. The results show great potential for GaN smart power integrated circuit (IC) application.

## 1. Introduction

GaN-based power devices are promising for high temperature, high frequency and high-power applications owing to their high two-dimensional electron gas (2DEG) mobility and high breakdown voltage [1,2,3]. However, mostly, the logic controller and the driver modules for GaN power devices are still implemented with Si integrated circuits (ICs) [4,5]. As we know, the larger parasitic inductance by integration could lower the performance of the GaN device under high-frequency and increase the cost. Hence, it is very important to develop GaN-based ICs to fully exploit the advantages of GaN-based ICs. GaN complementary metal-oxide-semiconductor (CMOS) logic is challenging because it is hard to get high-performance p-channel GaN. The direct-coupled field-effect transistor (FET) logic (DCFL) ICs with monolithically integrated E/D-mode n-channel devices offer a straight-forward and convenient approach to implementing GaN digital ICs.

Some methods based on D-mode GaN technology, for example, gate recess process [6,7,8] and F ion implantation [9,10] have been proposed to realize E-mode GaN high-electron-mobility transistors (HEMTs) and E/D-mode inverters. These works are either based on Schottky-gate HEMTs or MIS (metal-insulated-semiconductor)-gate HEMTs, and neither of them have been adopted in commercial GaN power devices, due to small gate swing in Schottky-gate HEMTs and gate-dielectric reliability concerns in MIS-HEMTs. The p-GaN gate technology has been approved to decrease the gate leakage and increase the gate voltage swing. It is already commercially available for its excellent characteristics and stability [2,11,12]. Some GaN ICs based on the p-GaN gate technology platform [13,14] have been reported in recent years; however, the systematic study and research of the E/D-mode GaN inverters based on p-GaN gate technology is not reported until now.

In this paper, monolithic integration of E/D-mode GaN inverters are demonstrated based on a p-GaN gate technology platform and the direct current (DC) and transient characteristics are studied. A very distinct property of GaN compared to silicon is its high temperature operation, and the characteristics of the GaN inverters operating at high temperatures are also studied in details. The inverters show large input voltage swing and wide noise margin from room temperature to 200 °C.

## 2. Materials and Methods

The schematic and the image of the E/D-mode GaN inverter are demonstrated in Figure 1.

The key processing steps of the E/D-mode GaN monolithic integration technology are shown in Figure 2. The E/D-mode GaN inverters were fabricated on p-GaN/AlGaN/GaN epitaxial layer on a 150-mm Si substrate. The epitaxial layer was grown through metal organic chemical vapor deposition (MOCVD) and included about 200-nm AlN nucleating layer, about 3.5-μm high resistive buffer layer, about 300-nm un-doped GaN channel layer, and about 18-nm Al_0.13_Ga_0.87_N barrier layer. Then, about 90-nm p-type GaN layer was grown and the Mg concentration was about 1 × 10^19^ cm^−3^. Finally, it was annealed within the MOCVD chamber at 650 °C for 15 min.

The E-mode and D-mode GaN HEMT were fabricated in the same batch. For E/D-mode GaN devices and inverters’ fabrication, the mesa region was formed with N ion implantation. Then p-GaN etching was carried out by BCl_3_/Cl_2_-based inductive couple plasma (ICP) with low DC power for low damage and high etching selectivity with Al_0.13_Ga_0.87_N. In this step, the p-GaN was totally etched for D-mode GaN HEMT and was partly left as p-GaN gate for the E-mode GaN HEMT. Then, plasma-free SiO_2_ chemical vapor deposition (CVD) was deposited as the first passivation. Au-free ohmic contact metal stack Ti/Al/Ti/TiN was sputtered after SiO_2_ etching. The ohmic contact metal was patterned by BCl_3_/Cl_2_ dry etching process. After that, the sample was annealed at 850 °C for 30 s in N_2_ ambient and the ohmic contact resistance was 0.6 Ω·mm. After removing SiO_2_ by SF_6_ plasma etching, about 20-nm atom layer deposition (ALD) Al_2_O_3_ was deposited at 200 °C as the gate dielectric for the D-mode device. Then, TiN was deposited and defined by ICP etching as the gate metal for D-mode GaN HEMT. For the E-mode GaN device, the gate window was opened by etching Al_2_O_3_ and SiO_2_ with low DC bias and Ti/Al gate metal was deposited and defined by ICP etching. Lastly, about 100-nm plasma enhancement chemical vapor deposition (PECVD) was deposited as the last passivation, and then the pad contact was opened by wet etching. D-mode and E-mode have a gate length of 1.5-μm and 1.8-μm, respectively. Both the D-mode and E-mode have the same gate width of 100-μm, gate-to-source distance of 1.5-μm, and gate-to-drain distance of 5-μm. *I-V* measurement was performed using Aglient B1500 semiconductor device analyzer.

## 3. Results

### 3.1. E/D-Mode GaN HEMTs

The DC output and transfer characteristics of E/D-mode HEMTs fabricated on the same wafer are plotted in Figure 3 and Figure 4. At room temperature (RT), the threshold voltage V_TH_ is +1.2 V and −14.5 V for E-mode and D-mode GaN HEMT by linear extraction method, respectively. The maximum drain current density I_DS,max_ is 312 mA/mm (at V_GS_ = 7 V) for the E-mode HEMT and 378 mA/mm (at V_GS_ = 2 V) for the D-mode HEMT. The maximum current density for D-mode HEMT is lower than normal devices should be, as a result of the lower Al concentration of the AlGaN barrier layer. As the limitation of the probe, the highest test temperature is 200 °C in the paper. At 200 °C, the V_Th_ of the E-mode HEMT is 1.1 V and with 0.1 V drift. The E-mode HEMT exhibits a maximum current density of 152 mA/mm at 200 °C, which is about 52% lower than that at room temperature. The lower maximum current density at high temperature may be due to the low mobility of the 2DEG, caused by impurity scattering [13,14]. For the D-mode HEMT the *V*_TH_ increases to −12.4 V, with +2.1 V drifting at 200 °C. The larger V_TH_ shifting for the D-mode HEMT at high temperature may be due to the fact that the 2DEG is captured by the interfaces of the gate metal/Al_2_O_3_/AlGaN structure [15]. The maximum current density of the D-mode exhibits 64% lower and it is 179 mA/mm.

Figure 4 plots the gate leakage characteristics of the E-mode and D-mode HEMTs. Both of them can operate safely at −6 V ~ +6 V at room temperature and 200 °C. The off-state gate leakage current increases about two orders of magnitude with the increase of temperature for the E-mode HEMT. The D-mode HEMT shows much lower gate current leakage at the forward bias and this may due to the high insulation quality of ALD Al_2_O_3_ layer.

### 3.2. E/D GaN Inverters

As shown in Figure 1a, the D-mode HEMT serves an active load and the E-mode HEMT serves as a driver of the inverter. In this paper, the gate-source-shorted D-mode HEMT is with a *W_G_/L_G_* = (14-μm/1.5-μm), and different driver-to-load resistance ratio α *= (W/L)_E_/(W/L)_D_* is fabricated with different *(W/L)_E_*. Figure 5 shows a typical E/D-mode GaN inverter with a driver/load ratio α *=* 15 and a supply voltage V_DD_ = 3 V. The red curve is the same transfer curve with the axis interchanged and represents the input-output characteristics. The output high (V_OH_) and low (V_OL_) voltage of the inverter are 3.0 V and 0.09 V, respectively, yielding a large voltage swing of 2.91 V (97%), close to the best values in the literature [10] with α *=* 64. The input logic low (V_IL_) and high (V_IH_) voltage level, extracted from the voltage unit gain points (G = *d*V_Out_/*d*V_in_ = −1), are 0.87 V and 1.33 V, respectively. The transition voltage region V_TR_ (defined as V_TR_ = V_IH_−V_IL_) is 0.46 V (15%), close to value as that in literature [10] with α *=* 64, showing the high performance of the logic inverter. The threshold of the inverter is 1.10 V (V_TH,inverter_, defined by V_out_ = V_in_). The logic low noise margin (N_ML_ = V_IL_ − *V*_OL_) is 0.78 V and the high noise margin (N_MH_ = V_OH_ − V_IH_) is 1.67 V.

Figure 6 shows the measured voltage transfer characteristics for E/D-mode GaN inverters of different α with V_DD_ = 3 V. Table 1 summarizes the parameters of the inverters.

As shown in Figure 6, the *V*_OH_ is 3.0 V with all α, indicating that the E-mode HEMTs are well switched off. The V_OL_ is decreasing with larger α and it is only 0.05 V with α = 28. The V_OL_ can be described as below.
(1)$$V_{\mathrm{OL}}=\frac{R_{E}}{R_{E}+R_{D}}V_{DD}\;$$

When the values of α increase, the R_E_ becomes smaller and equivalent resistance of E-mode transistor is much smaller than the overall resistance of the circuit, and the V_OL_ decreases. Also, larger values of α result in sharper transitions and smaller transition voltage. As a result, the output swing increases from 2.83 V to 2.95 V. As α increases from 7.5 to 28, the threshold of the inverter decreases from 1.30 V to 1.11 V, and the DC voltage gain (G) in the linear region increases from 5.5 to 8.3. Table 1 lists the measured values of static noise margins, as well as V_OH,_ V_OL_, output logic swing, V_TH, inverter_, and G. Both N*_M_*_L_ and N*_MH_* are improved as α increases. However, on the other hand, as the active load D-mode HEMT is kept unchanged and the gate length of the E-mode is kept as a constant, larger α will increase the capacitance of the E-mode HEMT, and this will decrease the speed of the inverter.

The static voltage transfer curves of the inverter with α = 15 were measured at different supply voltages and are plotted in Figure 7. The circuit performance parameters are listed in Table 2. When supply voltage increases, all the parameters of the E/D inverter increase accordingly. This means that the increase of supply voltage improves the static performance of the E/D inverter. Furthermore, magnitude of increase for the high noise margin is larger than that of the low noise.

High-temperature performance of the inverter circuits is characterized by varying the temperature from 25 °C to 200 °C in Figure 8. The insert figure plots the DC output and transfer characteristics of the inverter at 200 °C. Table 3 summarizes the parameters of the inverters at different temperature with V_DD_ = 3 V and α = 15.

As shown in the insert figure in Figure 8, the inverter functions well at V_DD_ of 3 V in terms of V_OH_, V_OL_, logic voltage swing, V_TH_, N*_M_*_L_ and N*_M_*_H_, and they are 3 V, 0.11 V, 2.89 V, 1.09 V, 0.68 V and 1.72 V, respectively. It should be noted that the V_OH_ of 3 V and V_OL_ of 0.11 V suggest that the inverter can successfully turn-on and turn-off the E-mode device even at 200 °C.Small variations of the inverter from RT to 200 °C can be ascribed to two factors. Firstly, the E-mode HEMT exhibits with a small variation of about 0.1 V at RT and 200 °C. Similar behavior of the threshold voltage was also demonstrated in [10]. Secondly, the low off-state leakage of the E-mode HEMT up to 200 °C is critical for the inverter to obtain a high level V_OH_ and low level V_OL_ at high temperature, making it possible to obtain a large logic voltage swing and noise margins without obvious variations for the inverter in such temperature range. The two factors work together to make this E/D-mode inverter capable of functioning properly up to 200 °C and exhibit small variations in terms of logic voltage swing, V_TH_, N*_M_*_L_ and N*_M_*_H_ from RT to 200 °C.

Figure 9 shows the transient output waveforms of the fabricated E/D-mode GaN inverters of different α with input of 20 kHz square wave from 0 to 3 V, and an edge time of 10 ns. As α increases from 7.5 to 28, the rising edge time of the output signal increases from 1.05 μs to 1.66 μs. The data are much smaller than that in the literature [15] and this may result from the excellent characteristics of the p-GaN gate E-mode HEMT. Generally, an inverter with smaller α may have poorer DC performance, but it will have better performance for dynamic characteristics. Designers will trade-off between the static and dynamic performance and select a proper α.

## 4. Conclusions

In this paper, AlGaN/GaN E/D-mode HEMTs and inverters were successfully fabricated on a 150-mm Si wafer based on p-GaN gate technology. The fabricated E/D-mode devices and inverters were characterized and analyzed. The inverter shows large input logic swing voltage, high noise margin and small transition voltage region. Excellent characteristics were displayed at high temperature, and small variations at high temperature in terms of logic voltage. The inverters also demonstrate small rising edge time of the output signal, and the results show great potential for GaN smart power ICs.

## Figures and Tables

**Figure 1 micromachines-12-00617-f001:**
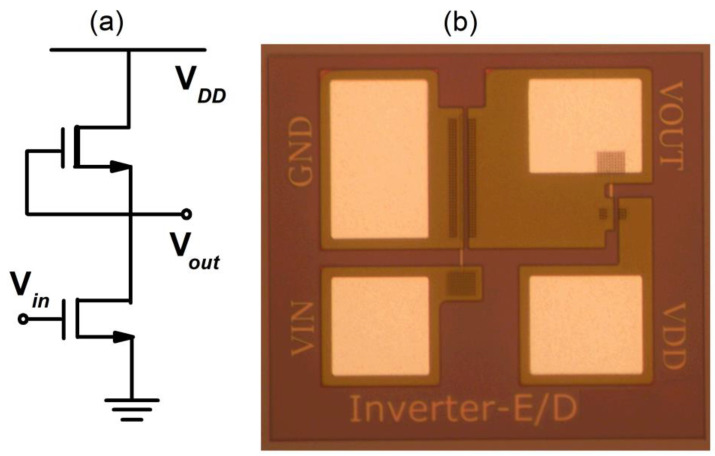
(**a**) A typical circuit schematic of the E/D-mode GaN inverter and (**b**) top view of the fabricated E/D-mode inverter.

**Figure 2 micromachines-12-00617-f002:**
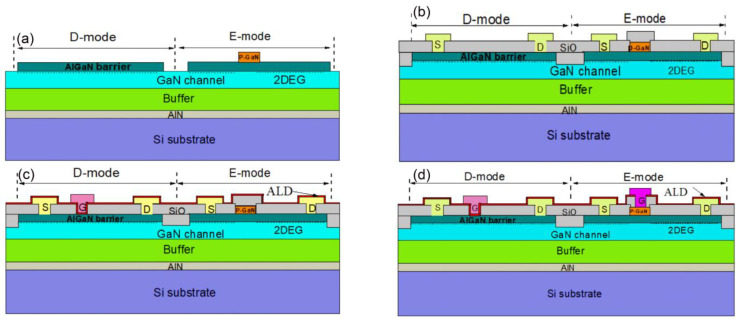
Schematic cross-section of (**a**) after p-GaN etching, (**b**) after source and drain formation and (**c**) after gate metal etching of the D-mode GaN high-electron-mobility transistor (HEMT) device and (**d**) after the gate metal etching of the E-mode GaN HEMT device.

**Figure 3 micromachines-12-00617-f003:**
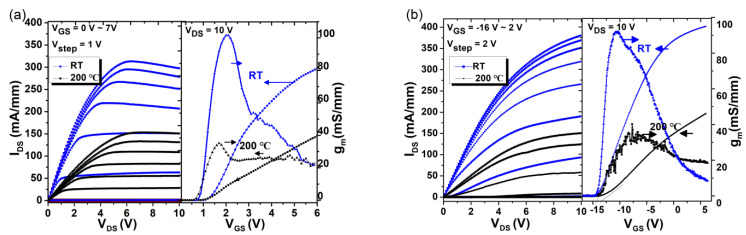
Direct current (DC) output and transfer characteristics for (**a**) E-mode GaN HEMT with L_G_/L_GS_/L_GD_ = 1.8/1/5-μm and (**b**) D-mode GaN HEMT with L_G_/L_GS_/L_GD_ = 1.5/1/5-μm.

**Figure 4 micromachines-12-00617-f004:**
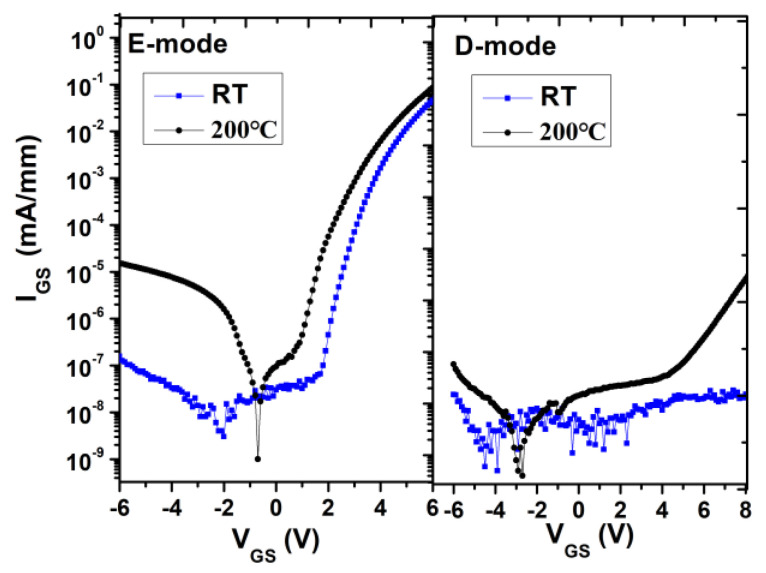
Gate leakage current for both E-mode and D-mode at room temperature and 200 °C.

**Figure 5 micromachines-12-00617-f005:**
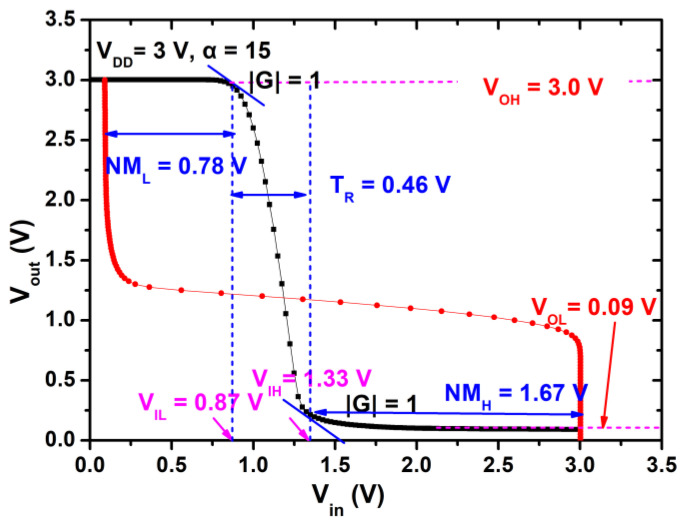
Static voltage transfer characteristics of the inverter with α = 15 measured at supply voltage V_DD_ = 3 V.

**Figure 6 micromachines-12-00617-f006:**
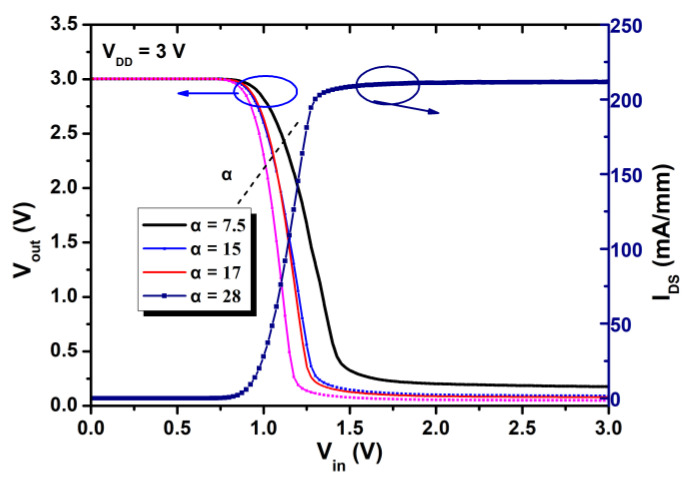
The static voltage transfer curve of the inverter with different α at V_DD_ = 3V.

**Figure 7 micromachines-12-00617-f007:**
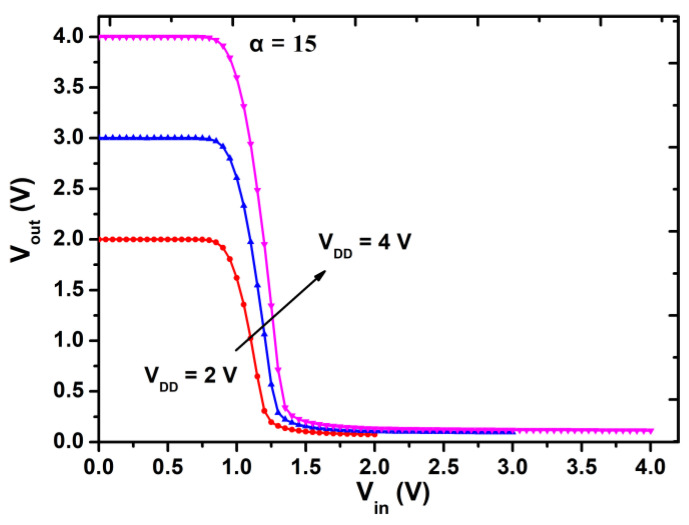
The static voltage transfer characteristics of the inverter with α = 15 measured at supply voltage V_DD_ = 2, 3, 4 V.

**Figure 8 micromachines-12-00617-f008:**
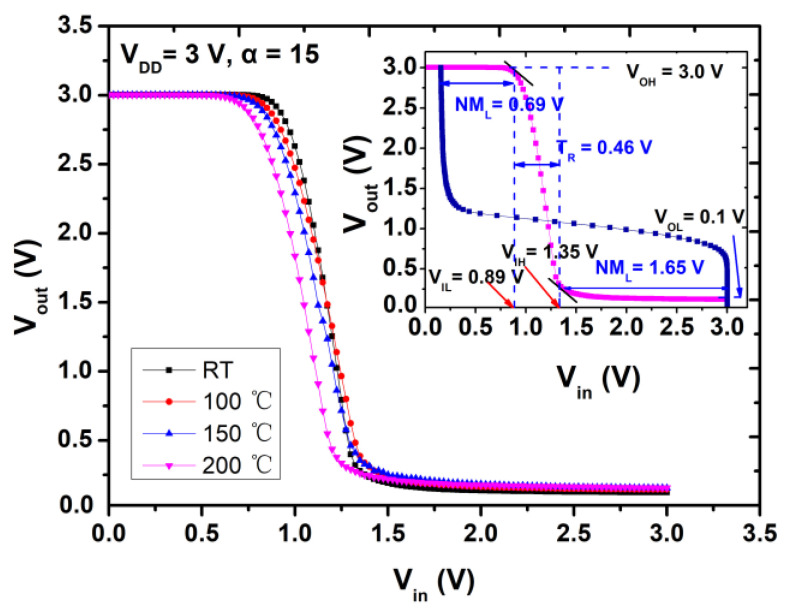
The static voltage transfer curve of the inverter at different temperature from 25 °C to 200 °C with V_DD_ = 3 V; insert: DC output and transfer characteristics of the inverter at 200 °C.

**Figure 9 micromachines-12-00617-f009:**
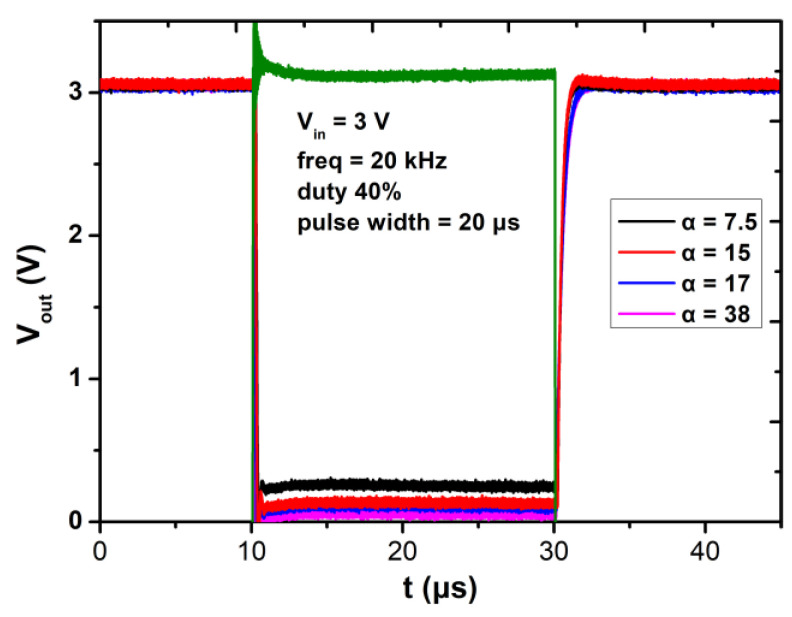
The transient output waveforms with input of 20 kHz square wave from 0 to 3 V with different *α*.

**Table 1 micromachines-12-00617-t001:** Noise margins for inverters with different α.

α	V_OH_ (V)	V_OL_ (V)	Output Swing (V)	V_TH_ (V)	G	N_ML_ (V)	N_MH_ (V)
7.5	3.0	0.17	2.83	1.30	5.5	0.76	1.53
15	3.0	0.09	2.91	1.18	6.8	0.78	1.67
17	3.0	0.07	2.93	1.17	6.9	0.79	1.70
28	3.0	0.05	2.95	1.11	8.3	0.80	1.80

**Table 2 micromachines-12-00617-t002:** Noise margins for inverters with different *V*_DD_ with α = 15.

*V*_DD_ (V)	*V*_OH_ (V)	*V*_OL_ (V)	Output Swing (V)	*V*_TH_ (V)	*G*	*N*_ML_ (V)	*N*_MH_ (V)
2	2.0	0.07	1.93	1.08	5.2	0.77	0.75
3	3.0	0.09	2.91	1.18	6.8	0.78	1.63
4	4.0	0.11	3.89	1.25	7.1	0.80	2.59

**Table 3 micromachines-12-00617-t003:** Noise margins for inverters at different temperature.

T (°C)	V_OH_ (V)	V_OL_ (V)	Output Swing (V)	V_TH_ (V)	G	N_ML_ (V)	N_MH_ (V)
*RT*	3.0	0.09	2.91	1.18	6.8	0.78	1.67
100	3.0	0.09	2.91	1.21	5.8	0.80	1.63
150	3.0	0.12	2.88	1.17	6.9	0.67	1.58
200	3.0	0.11	2.89	1.09	6.8	0.68	1.72

## Data Availability

The data presented in this paper is available with request from the corresponding author.

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
