# Peer review of "E/D-Mode GaN Inverter on a 150-mm Si Wafer Based on p-GaN Gate E-Mode HEMT Technology"

_micromachines, 2021, doi:10.3390/mi12060617_

Round 1
Reviewer 1 Report
This paper presents the monolithic integration of E/D-mode GaN inverters, based on a p-GaN gate technology platform. The authors are required to address the following issues:
- The language needs to be improved, such as “Besides a very distinct property of GaN 46 compared to Silicon is its high temperature operation so characteristics of the GaN in-47 verters operating at high temperatures are also studied in details.”.
- E/D-mode GaN inverter has been published in many papers, what is the main contribution of this paper, compared to others? Just investigation are not enough for the publication in this journal.
Author Response
Thanks for the reviewer’s helpful instruction, we have corrected the corresponding problems in the manuscript and we would answer these questions one by one.
This paper presents the monolithic integration of E/D-mode GaN inverters, based on a p-GaN gate technology platform. The authors are required to address the following issues:
Please see the attachment
- The language needs to be improved, such as “Besides a very distinct property of GaN 46 compared to Silicon is its high temperature operation so characteristics of the GaN in-47 verters operating at high temperatures are also studied in details.”
Response: we have revised the total paper and it has been revised as “A very distinct property of GaN compared to Silicon is its high temperature operation and the characteristics of the GaN inverters operating at high temperatures are also studied in details”.
- E/D-mode GaN inverter has been published in many papers, what is the main contribution of this paper, compared to others? Just investigation are not enough for the publication in this journal.
Response:
Compared with traditional gate recess, F ion implantation technology, one of the main significance of this work is that the E/D-mode GaN inverter is based on the commercial p-GaN gate technology, and it is of great potential for GaN smart power ICs in the commercial application. Although some GaN ICs based on p-GaN gate technology have been studied but systematic research of the E/D-mode GaN inverters has not been reported until now. A systematic study of E/D-mode GaN inverters based on p-GaN technology has been reported with details in this paper.
We have revised the investigation, as the second paragraph has been revised as following:
Some methods based on D-mode GaN technical for example gate recess process [8-10], F ion implantation [11,12] have been proposed to realize E-mode GaN HEMTs and E/D-mode inverters. These works are either based on Schottky-gate HEMTs or MIS (metal-insulated-semiconductor) -gate HEMTs and both of them have not been adopted in commercial GaN power devices, due to small gate swing in Schottky-gate HEMTs and gate-dielectric reliability concerns in MIS-HEMTs. The p-GaN gate technology can decrease the gate leakage and increase the gate voltage swing and it is commercially available for its excellent characteristics and stability [2, 4, 5]. Some GaN ICs based on the p-GaN gate technology platform[13,14] are reported recent years, however the systematic study of the E/D-mode GaN inverters based on p-GaN gate technology is not reported until now.

Reviewer 2 Report
The authors present an interesting work with nice results. Integration on GaN is of great interest today. The paper is structured well and written with average quality, however, it seems that there are mistakes in wording or missing words in some sentences.
Measurements were performed using Agilent B1500, correct? Maybe figures 3 and 4 can be put side by side with the same scale for better comparison. Gate leakage currents in figure 5 as well as transfer characteristics were measured only for the single sweep. However, hysteresis is typically observed in GaN transistor due to trapping effects. It is possible to show measurement in fig 3, 4 and 5 with a double stair sweep?
Only static voltage transfer characteristics are presented. The topic of GaN integration is studied for some time and therefore presenting only static characteristics for the system that is designed for switching decreases the significance and soundness of the paper. Please add switching characteristics that will prove that the inverter works also for higher frequencies.
As a general comment on the figures, the font size could be adjusted to be the same on all plots.
In the funding section, three numbers are at the end where the last one seems to be an artefact from editing.
Author Response
Thanks for the reviewer’s helpful instruction, we have corrected the corresponding problems in the manuscript and we would answer these questions one by one.
The authors present an interesting work with nice results. Integration on GaN is of great interest today. The paper is structured well and written with average quality, however, it seems that there are mistakes in wording or missing words in some sentences.
- Measurements were performed using Agilent B1500, correct? Maybe figures 3 and 4 can be put side by side with the same scale for better comparison. Gate leakage currents in figure 5 as well as transfer characteristics were measured only for the single sweep. However, hysteresis is typically observed in GaN transistor due to trapping effects. It is possible to show measurement in fig 3, 4 and 5 with a double stair sweep?
Response: Thanks for the suggestion. The measurements were performed using Agilent B1500 semiconductor device analyzer. Figures 3 and 4 have been put side by side with the same scale.
We tried to add the data for the double stair sweep, but unfortunately our heater for probe is out of order these days, and double stair sweep date at 200℃ is not available in a short time. The RT data is as following.
- (b)
Figure.1 DC output characteristics for (a) E-mode GaN HEMT with LG/LGS/LGD= 1.8/1/5μm and D-mode GaN HEMT with LG/LGS/LGD= 1.5/1/5μm
- (b)
Figure .2 DC transfer characteristics for (a) E-mode GaN HEMT with LG/LGS/LGD= 1.8/1/5μm and D-mode GaN HEMT with LG/LGS/LGD= 1.5/1/5μm
- Only static voltage transfer characteristics are presented. The topic of GaN integration is studied for some time and therefore presenting only static characteristics for the system that is designed for switching decreases the significance and soundness of the pap cies.
Response: Thanks for the constructive suggestion. We add the transient output characteristics in the paper. Figure 9 shows the measured transient output waveforms of the fabricated E/D-mode GaN inverters of different α with input of 20-kHz square wave from 0 to 3 V. The edge time is 10 ns and the rising edge time of the output signal is 1.05 μs to 1.66 μs. Further switching characteristics such as oscillator based the E/D inverter will be studied in our next work.
- As a general comment on the figures, the font size could be adjusted to be the same on all plots.
Response: the font size of figure 2, 3, 4 and 5 is adjusted to be the same with other figures.
- In the funding section, three numbers are at the end where the last one seems to be an artefact from editing.
Response: We have revised the last one funding number as “National Natural Science Foundation of China under Grant 61874114, 61674143 and 61974137”.
Please see the attachment

Round 2
Reviewer 1 Report
The authors dealt with my concern.
Reviewer 2 Report
Nice paper. The authors performed all requested corrections and added requested graphs.